



# A consistent dataset for the net income distribution for 190 countries, aggregated to 32 geographical regions and the world from 1958-2015

Kanishka B. Narayan[1], Brian C. O'Neill[1], Stephanie Waldhoff[1] and Claudia Tebaldi[1]

[1]Joint Global Change Research Institute (JGCRI), Pacific Northwest National Lab (PNNL)

*Correspondence to*: Kanishka B. Narayan (kanishka.narayan@pnnl.gov)

## Abstract

Data on income distributions within and across countries are becoming increasingly important to inform analysis of income inequality and to understand the distributional consequences of climate change. While datasets on income distribution collected from household surveys are available for multiple countries, these datasets often do not represent the same income concept and therefore make comparisons across countries, over time and across datasets difficult. Here, we present a consistent dataset of income distributions across 190 countries from 1958 to 2015 measured in terms of net income. We complement the observed values in this dataset with values imputed from a summary measure of the income distribution, specifically the GINI coefficient. For the imputation, we use a recently developed principal components-based approach that shows an excellent fit to data on income distributions compared to other approaches. We also present another version of this dataset aggregated from the country level to 32 geographical regions and the world as a whole. Our aggregation method takes into account both within-country and across-country income inequality when aggregating to the regional level. This dataset will enable more robust analysis of income distribution at multiple scales.

## 1. Introduction

Data on income distributions are important to understand trends in global and regional income inequality. These data are also routinely used to train models that project income distributions into the future (Fujimori et al., 2020; Hallegatte & Rozenberg, 2017; Hughes et al., 2009; Hughes, 2019; Soergel et al., 2021; Van der Mensbrugghe, 2015). In the climate literature, long-term projections of within-country income distribution have been used to inform analyses of how the impacts of climate change may affect inequality and poverty (Hallegatte & Rozenberg, 2017; Jafino et al., 2020). Income distribution data are generally collected through national and local household surveys. The most prominent sources of national-level income distribution data are the datasets presented by the World Bank through the PovCal tool (Bank, 2015) and the income distribution datasets available from the Luxembourg Income Study (LIS) (Ravallion, 2015; Smeeding & Grodner, 2000). Both these datasets present useful time series of income distribution for income groups such as deciles, based on multiple household surveys.

While these datasets have been widely used, they are subject to certain limitations. The definition of income in these datasets is often not the same, making comparisons across countries and





datasets difficult (Smeeding & Latner, 2015). For example, the PovCal dataset has mixed
observations for net income and consumption for the same country in different years. Such
inconsistencies can occur because the underlying surveys in different years might have been
conducted to measure different income concepts.  The two income concepts that these data tend
to use are:
i) ***Post tax income or disposable income or net income*** - This measure is defined as employee
income plus income from firms (self-employment) plus income from rentals (excluding any
payments), property income (these are generally capital gains and include dividends) plus current
transfers received (these include insurance benefits, employer contributions) less transfers paid
(taxes paid and employee contributions). This is the concept of income recommended by the
Canberra group for the international comparison of incomes (Europe, 2011).
ii) ***Consumption*** - This measure is the sum of food consumption plus non-food consumption plus
durable goods purchases (expenditure value minus cost of repairs) plus housing expenditures
(rent, mortgage payments) less any payments made (taxes, loan payments, asset purchases, etc).
This is the concept of income recommended by Deaton & Zaidi (2002) for welfare measurement.
Temporal and spatial coverage of the data are another issue. The LIS dataset provides consistent
data on the net income distribution. However, these data are only available for 50 countries from
1980 to 2016. The PovCal dataset provides data for a considerably higher number of countries
(165) compared to the LIS. However, the data are a combination of net income and
consumption-based observations (net income distribution data for 73 countries and consumption
distribution data for 118 countries).
Previous studies that have made use of these datasets for analysis or for modelling income
distributions have treated these income concepts as interchangeable (Rao et al., 2019; Sauer et
al., 2020). Moreover, for countries where no survey data on income distributions are available,
studies have used simple methods such as using a summary measure of income distribution such
as the GINI coefficient in combination with a parametric functional form such as a lognormal
distribution to impute the within country or within-region income distribution (Fujimori et al.,
2020; Rao et al., 2019; Shorrocks & Wan, 2008; Soergel et al., 2021).
There have been efforts to generate consistent datasets of the income distribution. However,
these efforts have been limited to local or regional data. For example, Frank (2009) generated a
consistent dataset of income distribution metrics for a single income concept for the fifty US
states. That particular study builds on previous studies that have compiled data for the US
states(Piketty & Saez, 2003). At the national level, there have been some efforts to produce
standardized datasets of income inequality, but they have generally been limited to summary
metrics of the income distribution such as the GINI coefficient (Babones & Alvarez-Rivadulla,
36     2007).

In this study we present a consistent dataset on national income distributions that represents a
single income concept namely, net income. This dataset is constructed by first choosing net
income decile data observations from all available sources for all available countries. For
countries that only have consumption distribution data, we impute the net income distribution



using a regression-based approach. For countries and years where no data on income distribution
is available, we impute income deciles using the GINI coefficient combined with a principal
component analysis (PCA) based method that provides a better fit to data than existing methods.
This PCA-based method was recently developed as a non-parametric approach to projecting
income distribution (Narayan et al., 2023).
One intended use of this dataset is to initialize income distribution variables in the Global
Change Analysis Model (GCAM) (Calvin et al., 2019). GCAM is a global, integrated model of
the energy, land, water, climate, and socioeconomic systems that produces projections for several
economic, climatological and physical systems variables for 32 geopolitical regions. Hence, we
also present income distributions for these 32 aggregated regions in addition to the 190 countries.
Our aggregation method takes into account cross-country inequality within a region in addition
to within-country inequality. Similarly, we also aggregated the country-level income
distributions for the world as a whole and show their temporal trends.
This dataset can be used to train projection models for income distribution across different scales
and, given the consistent income concept represented, can also be used to understand trends
within and across countries and regions. While these data are generated to enable modelling of
the income distributions in GCAM, they can be used to train any model for projecting income
distributions.
2. **Dataset construction**
We explain our approach for the dataset construction in detail in the sections below. To
summarize, we used the following steps:
a.  We first identified observations by country and year of net income deciles from all
23          available datasets (LIS, PovCal, and individual research studies). In doing so, we
24          prioritized the LIS dataset over all other datasets given its high data quality on the net
25          income distribution. Our selection process is explained in **section 2.1 and 2.2** below.
b.  For countries/years in which there were no net income data, but consumption data was
27          available, the net income distribution was imputed from the consumption distribution
28          using a regression-based approach. This is explained in **section 2.3**.
c.  Where there were no net income or consumption data, but the GINI coefficient, a
30          summary metric of the income distribution, i.e., was available, we imputed the net
31          income distribution from the summary measure using a PCA-based approach. This is
32          explained in **section 2.4**.


**2.1 Literature review and data selection from available household survey data**
We first conducted a literature review to identify sources of national-level data on income
distributions for as many countries as possible. There are three main datasets available, from the
Luxembourg Income Study (LIS)(Ravallion, 2015; Smeeding & Grodner, 2000) the World Bank
(whose data on income distributions are available through the PovCalNet tool) (Bank, 2015) and
UNU WIDER (which compiles data from different sources including the LIS, PovCal and other



research studies) (WIDER, 2008). Each dataset contains income distribution data for different
income concepts such as net income and consumption, based on nationally representative
surveys that may also represent sub-groups of the population (e.g., Urban vs Rural). These data
are sometimes supplemented with data from research studies, and they use different equivalence
scales to convert from household to per capita income. We first evaluated data availability for net
income deciles based on these criteria (income concept, scale, temporal coverage, and spatial
coverage).
In Table 1, we summarize these datasets differentiated by these criteria. Since the UNU WIDER
dataset is a compilation of data sources (i.e., LIS, PovCal or others), we also identified the
number of observations (country-year) in the UNU WIDER data derived from each source. **SI**
**Table 1** of this document summarizes some of the other studies which were used in the
collection of data for the UNU WIDER database. We are primarily interested in decile-level
income distributions derived from household surveys.

| Source | Income concept | Scale of survey | Countries | Years (range) | Observations (n) |
|---|---|---|---|---|---|
| Luxemburg income study | Net income | National | 50 | 1980-2016 | 347 |
| | Consumption | National | 25 | 1980-2016 | 209 |
| PovCalNet | Net Income | National | 73 | 1981-2018 | 1644 |
| | | Urban/Rural | 3 | 1981-2018 | 37 |
| | Consumption | National | 114 | 1981-2018 | 2341 |
| | | Urban/Rural | 3 | 1983-2018 | 54 |
| UNU WIDER | Net Income | National | 163 | 1979-2017 | 1707<br><br>347 from LIS<br>533 from other sources<br>827 from PovCal |
| | | Urban | 22 | 1961-2018 | 315<br><br>51 from PovCal<br>264 from other sources |



|  |  |  |  |  | 215 |
| --- | --- | --- | --- | --- | --- |
|  |  | Rural | 20 | 1950-2017 | 3 from PovCal<br>212 from other sources |
|  | Consumption | National | 66 | 1973-2018 | 1030<br><br>116 from LIS<br>779 from PovCal<br>135 from other sources |
|  |  | Urban | 5 | 1975-2017 | 52<br>45 from PovCal<br>7 from research studies |
|  |  | Rural | 5 | 1975-2017 | 50<br><br>46 from PovCal<br>4 from research studies |

*Table 1: Summary of coverage by data source*
We also evaluated access to microdata (i.e., underlying household-level data from household
surveys) for each of these datasets, since detailed microdata allows us to validate and understand
how the different income distributions for different income concepts were arrived at. Of all
datasets evaluated, we found that the LIS database has the most access to microdata via the
METIS tool (https://www.lisdatacenter.org/frontend).
The PovCal database maintained by the World Bank has the highest coverage geographically and
temporally in terms of observations. PovCal uses the disposable income data from LIS for high-
and middle-income countries and uses household survey data for consumption and disposable
income for low-income countries. The scales of the surveys are mostly national other than India,
China, and Indonesia where distribution data from separate rural and urban surveys are available.
Mean and median values of the income concepts are available in 2011 USD PPP converted using
country-specific conversion factors.
PovCal sometimes combines data of different types even within countries, e.g., for China,
PovCal uses income data in early years up to 1990 and then switches to consumption data.
Moreover, the micro-data for PovCal are not readily available.
UNU WIDER releases quality scores of individual datasets. It classifies the LIS database as
"High quality", due especially to the availability of metadata, and classifies the PovCal dataset as
"Average quality". Figure 1 below shows the income distributions by deciles for different
countries for different income concepts from the UNU-WIDER dataset.

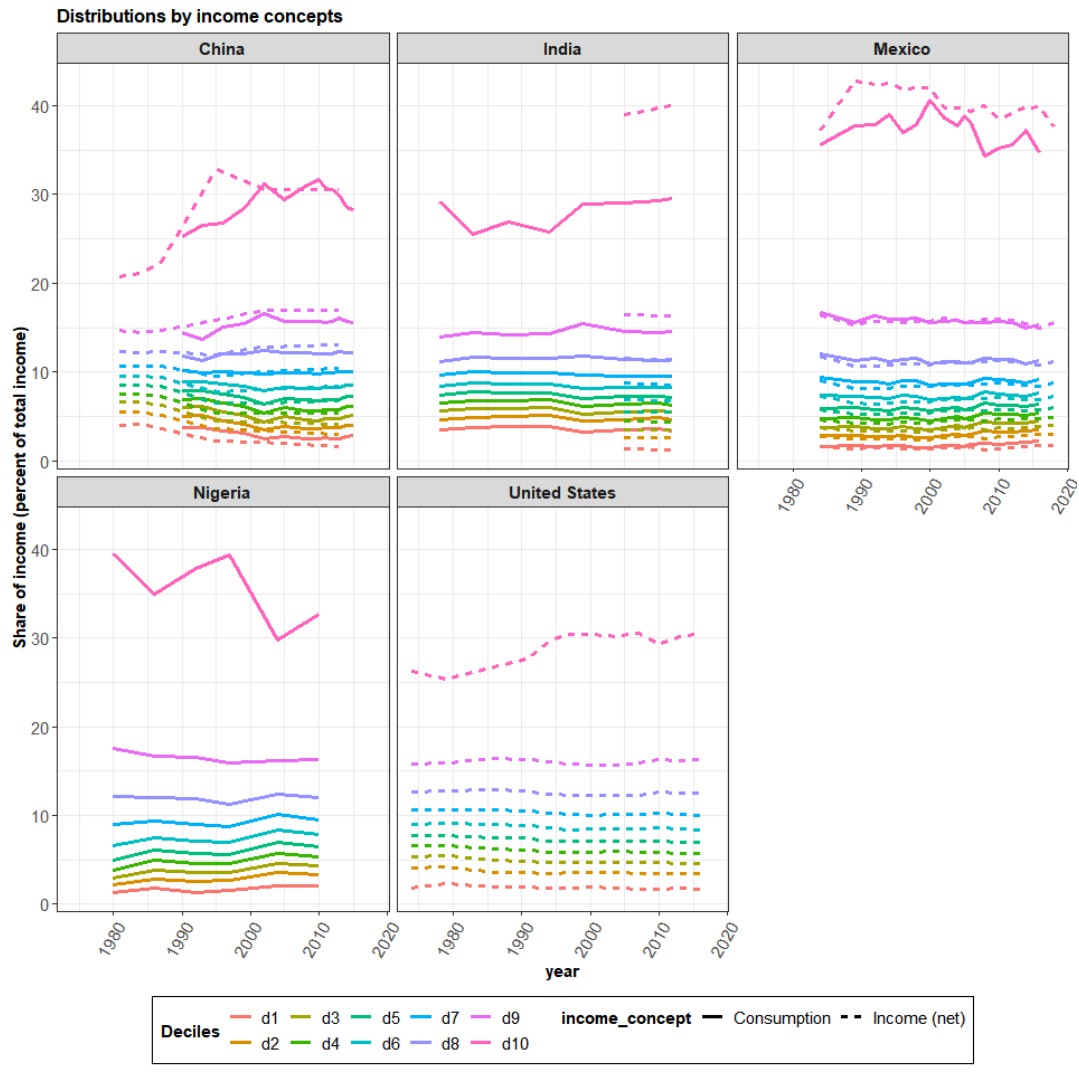

*Figure 1: Income distributions across countries (facets) for different deciles (color) for different income concepts (line types) from the UNU WIDER dataset*

**2.2 Selection of income concept and scheme for selection of data points**

We construct a dataset that represents solely net income based on the same per-capita equivalence scale. The per capita equivalence scale is calculated using total household income divided by the household size assuming equal sharing of income. Our process, summarized in Figure 1, improves upon other attempts to construct income distribution datasets from different sources (Rao & Min, 2018; Rao et al., 2019), since the previous studies used the income concept from different datasets interchangeably. We primarily select observations for net income deciles across countries from the LIS, given the high quality of data available from that dataset. We



begin by compiling separate datasets of the income distribution for net income and consumption.
In construction of both these datasets, we prioritize data points from the LIS. If no data were
available from the LIS for a country-year, we selected an observation of net income or
consumption from the PovCal database. Finally, if data were not available from that database, we
rely on income distribution data from other research studies available from the UNU WIDER
database. Note that when selecting values across multiple research studies we select values based
on the rating assigned by the UNU WIDER database to the studies. All data are selected for the
equivalence scale applied in the WIDER dataset, in which household income was converted to
per capita units by dividing the household income by the household size assuming equal sharing
of income.
Thus, at this stage, we compiled two different data sets, one that represents net income
distribution across countries across time and another that represents consumption for the same
countries. Now, we prioritize the selection of net income distribution values over consumption
for each country-year.
Where data are only available for the consumption distribution, we convert the consumption data
to net income data (as explained in section 2.3 below), using a regression approach to generate a
harmonized dataset of net income deciles. Where necessary, we aggregated data sources across
different survey scales (urban vs. rural) using a population-weighted average.
Figure 2 summarizes our data selection approach.







2   *Figure 2: Summary of data selection approach for each country, year observation*



Based on the above, we evaluated data coverage for the 229 countries we are targeting. The
geographical boundaries of the 32 GCAM regions are defined based on these 229 countries
(countries with their corresponding regions are listed in **SI Table 2**).  We identified observations
after the selection above for four categories, namely countries where we have net income data for
at least one year, countries where we had both net-income and consumption distribution data for
at least one year (in case of these countries we selected the net income distribution value for
deciles), countries where we had only consumption data, and countries where there were no data
(these countries only had data on aggregate measures of inequality such as the GINI coefficient
but no data on income deciles). Table 2 below summarizes the number of observations (country
years) by category of data.

| Data availability (for at least 1 year) by income concept | Number of countries | Notes on use |
|---|---:|---|
| Net income only | 33 | Use net income share data. |
| Both net income and consumption | 54 | Use net income share data. |
| Consumption only | 83 | Imputed income shares to be calculated (See section 2.3) |
| No decile data available but GINI is available | 14 | Impute deciles based on GINI coefficient (See section 2.4) |
| No data available | 39 | Drop from data set (section 5) |
| **Total** | **229** | |

*Table 2: Summary of data availability by income concept.*
**2.3 Imputing net income shares using consumption shares**
Using data for countries which had both income and consumption distribution observations for
the same years (n=257, across 54 countries where each of which have data for ten deciles of
consumption and the ten deciles of net income), we constructed linear regression equations for
each decile to impute the net income shares using the consumption shares of the income
distribution (Figure 3). The highest R squared value was observed for the fifth, sixth, seventh and
tenth deciles d10 of 0.74 and the lowest R squared value was observed for d9 of 0.37.



*Figure 3:Consumption distribution deciles (x axis) compared to Net income distribution deciles (y axis) across all country-year observations. Dashed lines show the 1:1 linear relationship. Solid line is the used regression line. Only observations for half the dataset are selected (Pre 2004) for the plot*

Consumption distribution deciles are converted into net income deciles using the equation (1)
below,
$$D_{netincome_{n,r,t}} = Coeff_n * D_{consumption_{n,r,t}} + Intercept_n \qquad (1)$$
where,





D is the share of consumption or income in a particular decile between 0 and 100,
Coeff is the coefficient applied to each decile parameterized using a linear regression,
documented in Table 3,
Intercept is derived from linear regressions run for each decile, documented in Table 3,
$n$ is the decile ranging from 1 to 10, and
$r, t$ are the region and the time step respectively.

| Decile | Intercept | Coefficient | Adjusted $R^2$ |
|--------|-----------|-------------|-----------|
| 1 | -0.02 | 0.81 | 0.5 |
| 2 | -0.39 | 1.00 | 0.64 |
| 3 | -0.65 | 1.06 | 0.69 |
| 4 | -0.76 | 1.08 | 0.72 |
| 5 | -0.91 | 1.10 | 0.75 |
| 6 | -1.12 | 1.12 | 0.78 |
| 7 | -1.10 | 1.10 | 0.78 |
| 8 | -0.74 | 1.06 | 0.66 |
| 9 | 4.81 | 0.69 | 0.29 |
| 10 | -1.39 | 1.11 | 0.75 |

*Table 3: Summary of coefficients and intercepts by decile used by Equation 1. These are fit*
*based on 257 data points.*
The final dataset therefore includes **8422** observations based on distributions of consumption or
net income across 170 countries spanning the time-period 1958-2018.
**2.4 Imputing net income deciles based on summary measures of the GINI coefficient.**
For many countries, years, no data are available for the income or consumption deciles based on
household survey data. However, World Development Indicators (WDI) dataset (Reid, 2012) do
provide aggregate measures of the income distribution such as the GINI coefficient for some
country-year observations[1]. Many studies have utilized the GINI coefficient in combination with
different functional forms to estimate the underlying income distribution (Shorrocks & Wan,
2008; Soergel et al., 2021). Most prominent amongst these methods is the usage of the lognormal
functional form along with the GINI coefficient to derive the underlying distribution.
However, methods such as the lognormal functional form have documented limitations. For
example, the observations are known to deviate from the lognormal in the tails of the
distribution(Badel et al., 2020; Chotikapanich, 2008). Moreover, the lognormal functional form

---

[1] The WDI dataset has observations of the GINI coefficient from various research studies. However, the underlying income concept of the GINI coefficient is not always available.



is assumed for every country for every year. Recently, a non-parametric approach was developed
which uses the GINI coefficient in combination with a two-component model based on a
principal components analysis (PCA) to produce a more accurate estimate of income deciles
(Narayan et al., 2023). This method addresses some of the limitations of the lognormal
functional form. The performance of the non-parametric PCA based approach compared to the
lognormal functional form is described in more detail in **SI 2 Figure 1**. For country-years where
we could not find data on net income or consumption, we used this PCA based approach along
with observed values of the GINI coefficient from the World Development Indicators Database
(Reid, 2012) to impute the underlying net income distribution.
The PCA based approach can be described as follows.
The income deciles are calculated as
$D_{r,t} = a_{r,t}PC1 + b_{r,t}PC2$        (2)
Where,
D is a 10-dimensional vector of income shares for all population deciles in region r at time t.
PC1 and PC2 are the two principal components, also vectors of length 10 (Values of PC1, PC2
are provided in **SI 2 Figure 2, SI 2 Table 3**)
a and b are coefficients of the two principal components specific to each region and time
The coefficient *a* is derived from the GINI coefficient using a regression equation estimated on
**1659** observations of national net income distribution
$a_{r,t} = -11.4815 + 29.71708 * GINI_{r,t}$        (3)
And the coefficient *b* is estimated using lagged values of the Palma Ratio (d10/(d1+d2+d3+d4))
and income share in the ninth decile and the current period labor share of GDP
$b_{r,t} = -17.18222 + (1.07957 * LabShareGDP_{r,t}) + (113.10810 * Ninth\ Decile_{t-1})$
$+ (-0.36392 * PalmaRatio_{r,t-1})$        (4)

Using this approach, we were able to fill in values for various country-years. The observations in
our dataset are now summarized in Table 4

| Type of data | country-year observations |
|---|---|
| Original data on net income | 1191 |





| | |
|---|---|
| Imputed based on original data on consumption | 394 |
| Imputed from GINI coefficient (using PCA algorithm) | 6837 |
| **Total** | **8422** |

*Table 4: Summary of observation types in final data set*

As observed in Table 4, the majority of observations in our dataset are those from the imputation from the GINI coefficient. We have classified each observation based on the categories above in the final dataset so that users can select any subset they prefer.

The PC algorithm used for the imputation was tested against the latest data on decile level income distributions and provided a good fit for all deciles across all countries. This testing was performed both for in sample and out of sample observations. This PCA based method was also found to yield a better fit to the data when compared to other methods such as using a GINI coefficient in combination with a lognormal functional form.

Since we used a summary measure (GINI coefficient) to derive the underlying distribution, we also validated our imputation approach by recalculating the GINI coefficient from the imputed distribution and comparing it with the reported GINI coefficient (Figure 4). We observe that our re-calculated values largely have a one-to-one correlation with the input GINI values suggesting that the imputation did not introduce many errors (overall R squared value of the comparison is 0.99). However, the relationship does start to weaken for countries with very high GINI coefficients such as South Africa where the recalculated GINI coefficient is different from the observed GINI coefficient by as much as 0.07 points. This is a result of the parameters of the PCA algorithm which do not reproduce well values for outlier countries with extreme GINI coefficients. We also observe that the re-calculated GINI coefficients for some countries are different in different years. For example, in Malawi, there are large year to year jumps in the reported GINI coefficients from year to year (**SI 2 Figure 3**).

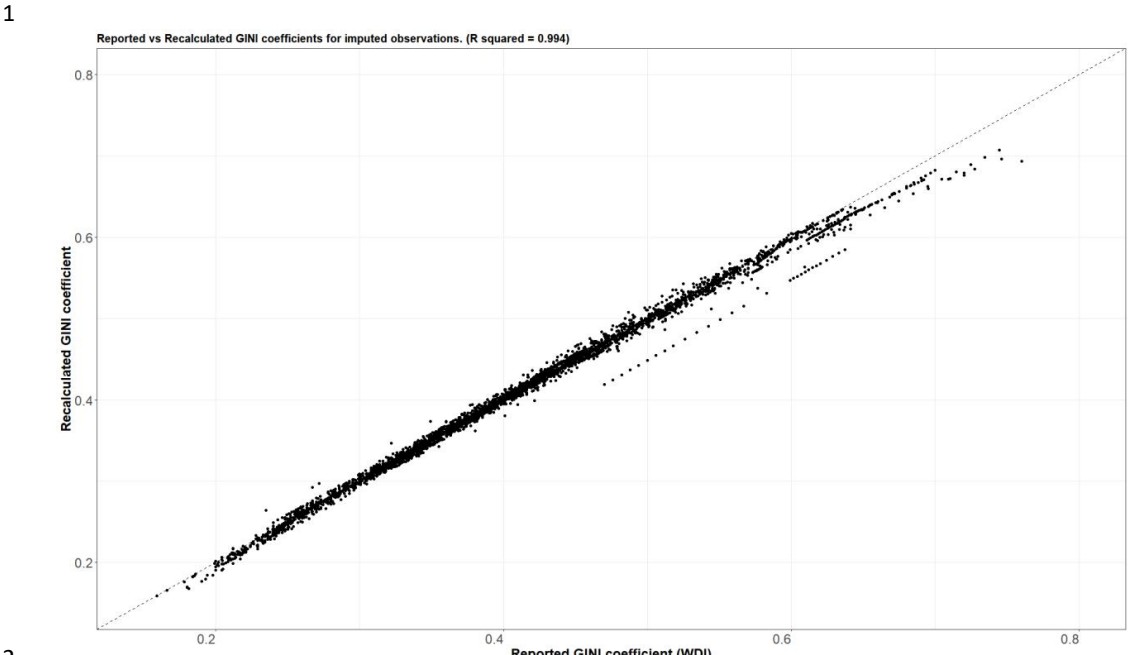

*Figure 4: Comparison of the reported GINI coefficients from the WDI (x axis) with the recalculated GINI coefficients from the imputed distribution (y axis). Each dot is a country-year observation. The dashed line represents a one-to-one relationship.*

We also evaluated temporal trends in the complete dataset which now include values from direct observations and also imputed values. The top two panels in Figure 5 below shows trends in the income shares for the 10[th] decile for India and China across time from all data sources.

This approach helps us generate better coverage in our dataset and the PCA model provides a statistically valid method to generate the data from GINI coefficients. This approach does have some limitations, however. In the United States for example, we observe that the imputed income distribution values are consistently higher than observed values in all years (with income shares in upper deciles being approximately 5% higher when imputed compared to the actual data). This is likely because the GINI for the US from the World Development Indicators database is based on gross income and the income distribution based on surveys (From LIS) is for net income, i.e., it includes adjustments for direct taxation[2]. This suggests a limitation in our imputation approach and one possible next step would be to only use net income GINIs for the imputation of the decile level income distribution. To implement this next step, we would require a dataset that clearly defines the income concept for the GINI coefficient provided. A good first step in this this direction would be to use data from the "All the GINIs" dataset which clearly specifies the income concept of the derived GINI coefficient (G. Ferreira et al., 2015; Smeeding & Latner, 2015).

---

[2] Note that the examination of the metadata for the LIS values for the US shows that the values are computed as the gross income distribution minus an imputed tax adjustment.





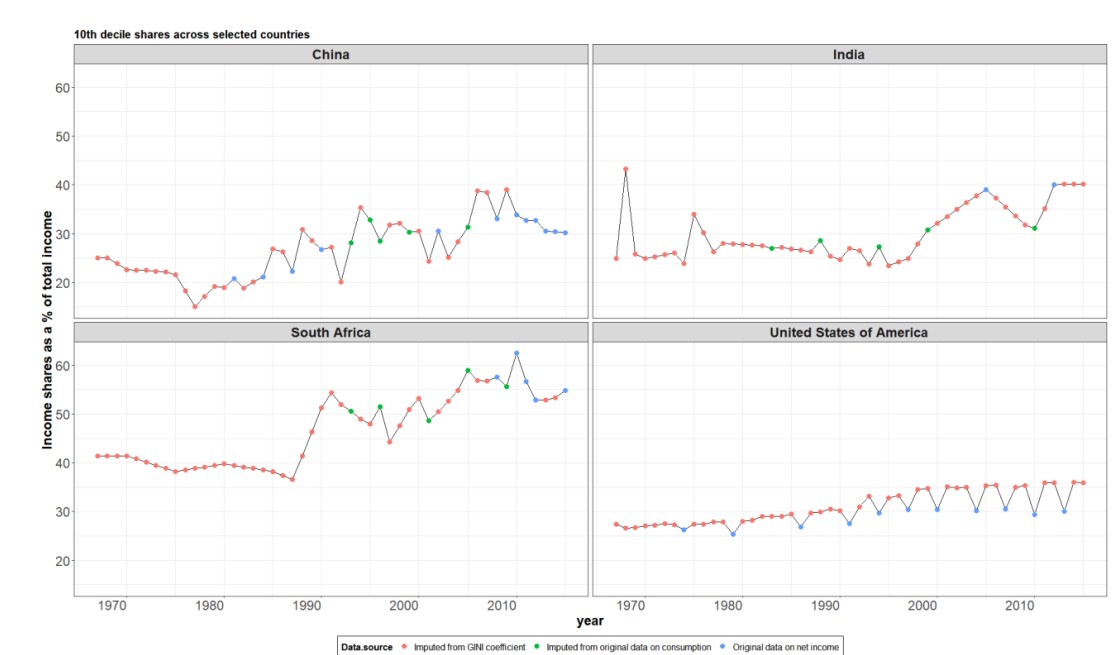

*Figure 5: Temporal trends in the 10th decile for the complete dataset. Colors represent different data sources.*
**3. Aggregating income distributions to the regional level**
The motivation for developing this country-level dataset was to initialize decile level income
distribution values for the Global Change Analysis Model (GCAM). As mentioned above, we
aggregated the income distributions from the country level to 32 geographic regions represented
by GCAM. The 32 regions are shown as a map in Figure 6.



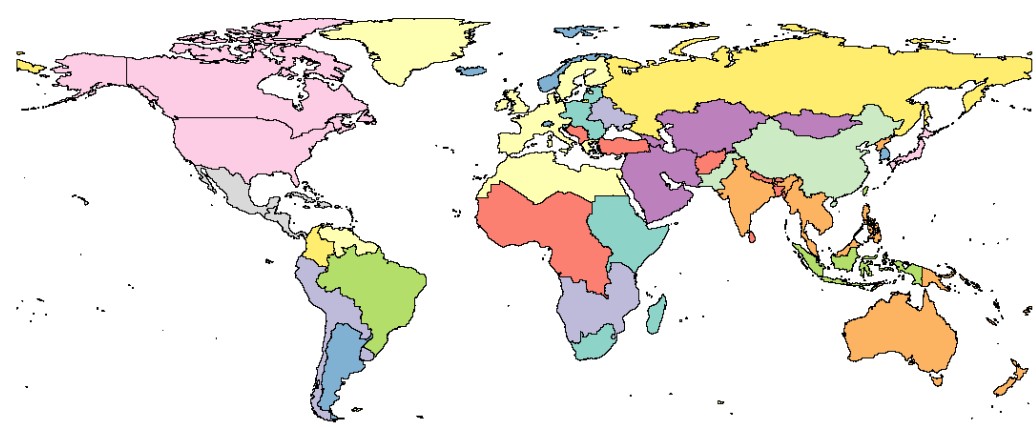

*Figure 6: Map of the 32 GCAM regions. These 32 GCAM regions are based on 229 country boundaries.*
Aggregating income distributions to the regional (where a region is made up of multiple
countries) level is not straightforward because countries within regions differ in population size,
average income level, and level of inequality in the income distribution. For example, an
individual who belongs to the 10th decile in Romania would not necessarily be counted amongst
the 10th decile of Europe as a whole, given the difference in the overall income level of Romania
relative to higher income level of other European countries such as Germany and France.
Similarly, even countries with similar average income levels may differ significantly in how
income is distributed across deciles.
The aggregation of the country level income distributions to the regional income distributions
involved the following steps:
1. First, we sorted all country net-income deciles in the region by the average decile income
level, from lowest to highest income (The net income distribution shares are applied to
this GDP per capita, measured in at PPP (2011 USD) to arrive at the income level).
2. Next, we calculated the cumulative population for each of these country income groups.
The cumulative population over all country income groups matches the regional total
population.
3. We then calculated cumulative population cutoffs that would create regional population
deciles by dividing the regional population by 10.
4. Based on these cutoffs, we calculated the regional decile shares of income by assuming a
uniform distribution of income within each country-decile. Thus, wherever a country
decile spanned a regional cutoff, its income was split between regional deciles in
proportion to the country population falling in each regional decile.
Figure 7 below illustrates our aggregation approach for GCAM region 14, Europe Non-EU,
which is made up of Albania, Bosnia, Croatia, Macedonia, Montenegro, Serbia and Turkey. The



figure demonstrates that a given regional decile can contain a mix of deciles at the country level.
For example, the regional d2 consists of d3 and d4 values of some low-income countries such as
Serbia and Albania. The regional d10 contains both the d9 and d10 values from Tukey.

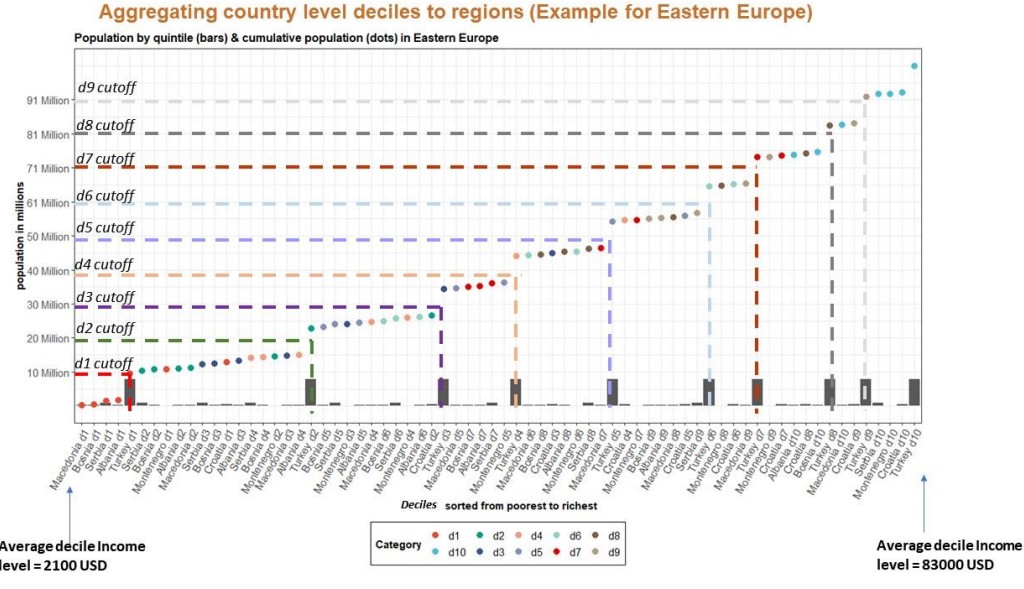

*Figure 7: Explanation of our aggregation approach. On the x axis all deciles within the region are sorted from low income to high*
*income. Bars track the population. The dots show the cumulative population compared to the decile level income. Dashed lines*
*show the new regional cutoffs for the deciles.*
We also compared the aggregated income distribution to the country level income distributions
for 2015 (Figure 8). We find that the aggregated income distributions are mostly driven by trends
in the income distribution of the most populous countries in the region, as expected. In the
example above, the income distribution for GCAM region 14 (Europe Non-EU) is largely driven
by the income distribution of Turkey, which is the most populous, and most unequal, country in
that region (e.g., Turkey represents approximately 75% of the regional population in 2015).
There are certain cases where the regional distribution is significantly different than the country-
level distributions. In Central Asia for example, the regional income distribution is much more
unequal (regional GINI is 0.53) compared to the country level GINIs (Highest GINI is 0.39).
This is because there is considerable variation in the income levels across countries. The
country-level average incomes range from USD 2011 in Tajikistan to USD 23485 in Uzbekistan.
This further illustrates why a specific aggregation method was necessary to construct these
regional income distributions (Simple aggregation methods would miss such intra-regional
dynamics).

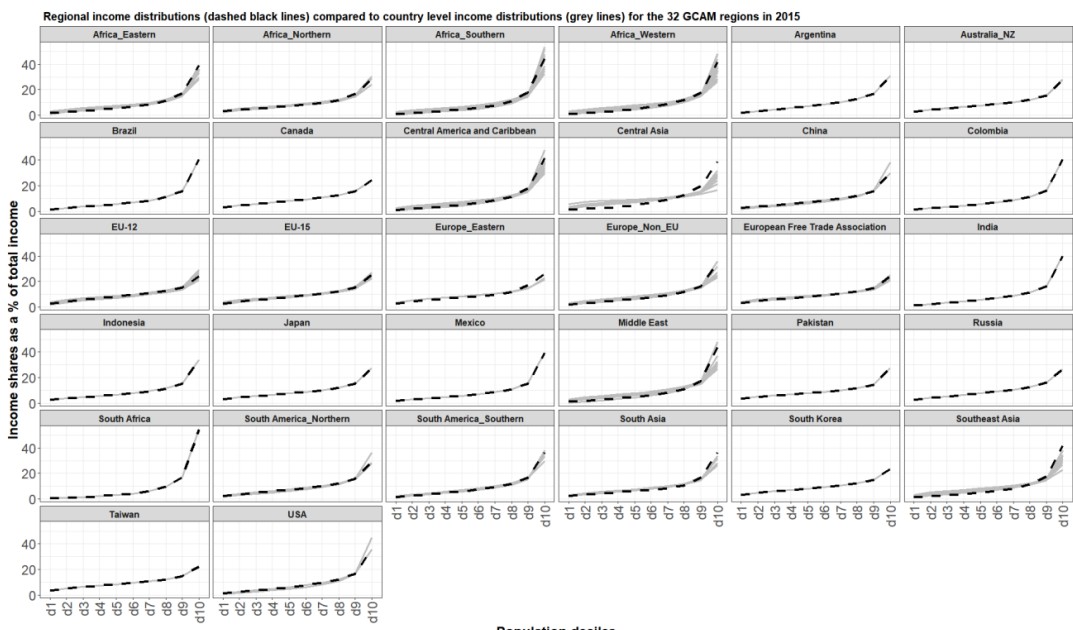

*Figure 8: Regional income distributions (Dashed black line) compared to the national income distributions (grey lines) in each of*
*the 32 regions in 2015.*
**4. Aggregating country income distributions to the global level**
Using the same aggregation methodology described in section 3, we also aggregated all the
country level distributions to the global level. This yielded a time series of income deciles for the
world as a whole. Figure 9 below shows the global income deciles relative to the country level
deciles in selected years. Changes in the global income distribution are largely driven by changes
in the country level distributions of high population countries. For example, China's income
distribution has become more equal since 2010 (The GINI coefficient of the country has reduced
from 0.42 to 0.38) and this is reflected in the changes in the global income distribution.
Inequality between countries has generally reduced with lower income countries growing in
income faster than high income countries. Figure 10 shows the evolution of the income
distribution between countries as a global GINI coefficient between 1990 and 2015. Our dataset
of income distributions at different scales would allow more robust analysis for multiscale
modelling of income distributions.

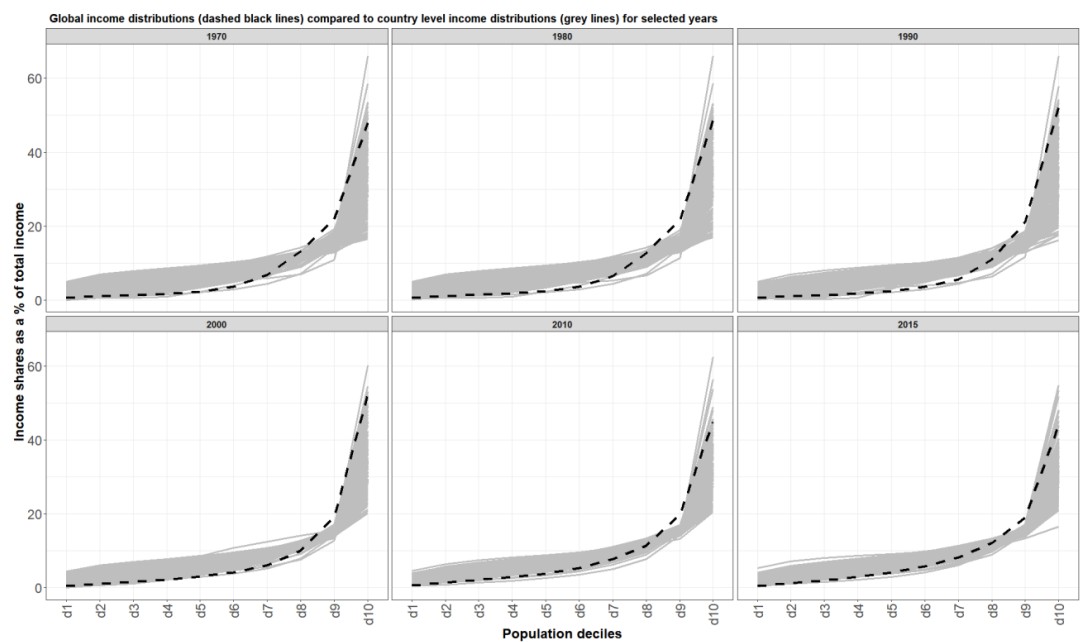

2  *Figure 9: Global income distributions (Dashed black line) and country level distributions (grey lines) for different years by decile.*

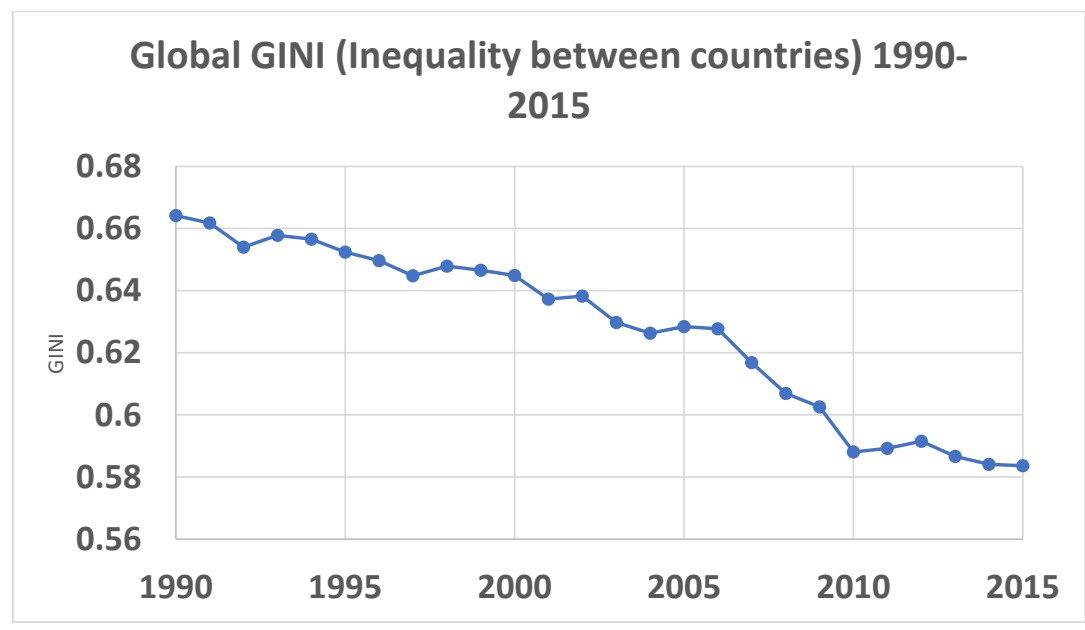

5  *Figure 10: Global GINI index over time. GINI is re-calculated from decile level distributions of net income.*



5. **Quantifying coverage and assessing regional bias in the data**
As mentioned earlier, we intended to develop a dataset for net income distribution for the 229
countries aggregated to 32 regions used in GCAM. As shown in Table 2, we were unable to find
any data on net income or consumption for 39 of those 229 countries. Previous models that have
been developed for projecting income distributions have been based largely on data for high
income countries (Rao et al., 2019; Sauer et al., 2020).
In order to evaluate whether the lack of data for the 56 countries introduces a bias, we assessed
the data coverage in terms of percent of global population (total population of 229 countries) and
percent of global GDP (total GDP at MER for 229 countries) for our dataset. We found that our
dataset covers 98% of the global population and 93% of the global GDP in any given year.
Similarly, we also compared the average population and GDP of countries with and without data
for five years (Table 5) and found that the average population of countries with data in the last
historical year, i.e., 2015, is significantly higher (19 times) than the average population of
countries without data. Similarly, the average GDP of countries with data is roughly 4.5 times
the average GDP of countries without data.

| | Average national population (in thousands) | | Average national GDP at MER (Billion 2010 USD) | |
|---|---|---|---|---|
| Year | Data available | Data not available | Data available | Data not available |
| 2010 | 37988 | 2835 | 370 | 90 |
| 2011 | 38881 | 2777 | 385 | 90 |
| 2012 | 39351 | 2808 | 394 | 90 |
| 2013 | 39822 | 2838 | 404 | 91 |
| 2014 | 40066 | 2915 | 414 | 91 |
| 2015 | 40610 | 2063 | 423 | 93 |

*Table 5: Comparison of national average population and national average GDP (at MER) for*
*countries with and without data for five historical years.*
Since this data will be used to initialize income distributions in the GCAM model, we also
evaluated whether the data would introduce a bias for any GCAM region (e.g., is there no
coverage or poor data coverage for any given GCAM region).
To evaluate this, we divided the countries in our dataset into the 32 geographical regions
modelled by GCAM. We then assessed the data coverage in terms of a percent of population (SI
3 Table 4) and GDP (SI 3 Table 5) for each of these regions. While these regions are specific to a
particular model, they also well represent heterogeneity across countries in terms of regional
economic and demographic conditions.
An example of a result of this assessment is that in the region of Africa Eastern we found data
that covers 64% of the region's population in 2010 and 40% of the region's GDP for the same
year. We performed this assessment for 5 years from 2010 to 2015. The purpose of this
assessment is to verify whether we have some coverage of data for all regions of the world





within those 5 years which would increase our confidence that our models are not biased towards
high income countries. The lowest coverage in our dataset is found for the Middle East region
where our data covers roughly 60% of the region's population and 40% of the region's GDP.

## 6. Discussion

In this paper we present a new consistent dataset on the net income distribution across 190
countries from 1958-2015. This dataset is also available for 32 aggregated regions and the world
as a whole. To our knowledge there is no other dataset that presents consistent data at multiple
geographical scales that has been documented in a peer-reviewed article. This complete and
harmonized dataset may be useful for efforts related t modelling of the net income distribution.
The aggregation method presented in this paper (section 4) takes into account both within-
country and across-country inequality when aggregating income distributions to regions or the
world. This is important to regions where there is significant diversity in the income distribution
across countries such as Central Asia, where the aggregated income distribution is significantly
more unequal than any of the member countries (Figure 8).
There are a number of areas of improvement that we have noted that can be explored as next
steps or in future updates to this dataset. First, we have used a simple linear regression approach
when converting the consumption distributions to net income distribution. This can be improved
upon if more data becomes available related to the savings rate across countries or if the income
within countries can be broken down into the various incomes and expenditures similar to a
Computable General Equilibrium (CGE) framework.
Similarly, while our imputation approach greatly increased spatio-temporal coverage in our
dataset, we also observed that in some cases, such as the US, the imputation approach generates
consistently higher inequality metrics compared to the original household survey data. This is
because the GINIs used from the WDI could be based on gross income, which overestimates
inequality based on net income. In the future, these gross income GINIs should also be converted
to net income GINIs before the imputation. This would require more detailed data on the input
GINI coefficients. One possible next step would be to re-conduct the imputation with GINI
values from datasets such as the "All the GINIs" dataset which tracks the type of the GINI
coefficient (G. Ferreira et al., 2015; Smeeding & Latner, 2015). Another option would be to
explicitly generate a tax adjustment to convert gross income values to net income.
We further found that the PCA based imputation approach generates some error when imputing
the income distributions of highly unequal regions such as South Africa. As more data on income
distributions becomes available, the PCA algorithm can be re-parameterized to newer data.
When this happens, the imputation should be re-performed.
Finally, the data generation described above is documented as an open-source workflow of a
software package called *pridr* which can be used to generate and re-aggregate these data. The
software package is available on GitHub and the dataset itself is available as a version-
controlled release on Zenodo (See data availability statement below).



## 7. Data availability

The main dataset is available here on Zenodo- https://zenodo.org/record/7093997 (Narayan et al. 2022) There are 3 main datasets available –

1. 32 region income deciles from 1958 to 2015
2. Global (Entire world as one single region) income distribution from 1958-2015
3. ISO level income distributions from 1958-2015

## Competing interests

The authors declare that none of the authors have any competing interests.

## Acknowledgements

This research was supported by the U.S. Department of Energy, Office of Science, as part of research in Multi Sector Dynamics, Earth and Environmental System Modeling Program. The Pacific Northwest National Laboratory is operated for DOE by Battelle Memorial Institute under contract DE-AC05-76RL01830

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
