# Peer review of "A consistent dataset for the net income distribution for"

_Earth System Science Data, 2023_

## Referee Comment (RC2)

**Review - A consistent dataset for the net income distribution for 190 countries, aggregated to 32 geographical regions and the world from 1958-2015**

The paper puts together a world income data set drawing on several global repositories and filling the data gaps with a regression and a Principal Component Analysis (PCA) approach. The authors obtain a world database for 190 countries and 32 geographical regions.

The paper contributes to a growing literature on world databases of incomes. The data sources selected are appropriate but insufficient. There is no mention of the work led by Branko Milanovic in the US and Thomas Piketty in France. Both these authors oversee global data efforts to measure income and income inequality that should be reviewed/acknowledged by the authors.

The central contributions of the paper are the regression-based approach to estimate net incomes from consumption data and the PCA approach used to estimate income data from national Gini coefficients.

The regression-based approach (section 2.3) is popular but there are better methods that should provide better fit including quantile regressions and random forest methods. The problem with simple OLS is that they are very poor at predicting incomes on the tails of a distribution (the predicted income distribution is always much narrower than the original income distribution). The fact that the authors run different regressions for each decile accentuates this problem by creating discontinuities between deciles. This problem can be overcome with quantile regressions or, better, with random forest. The probabilistic nature of random forest fits the tails if income distributions much better than standard OLS. I would recommend the authors to test both methods and compare results with the current ones.

The PCA method (section 3.4) is somewhat unconventional for this specific literature. This, per se, is not a critique, but it does require validation beyond what the authors offer. Here I would suggest taking the entire net income distribution for a few countries where these data are publicly available, calculate the net income deciles and the Gini coefficient, plug this Gini into equations 3) and 4) and compare the resulting estimated deciles with those calculated from the full data. Also, it is important to clarify where the coefficients in equations 3 and 4 come from. I could not find the model and the results of the "equation estimated on 1659 observations".

The revisions suggested above are substantial and I would recommend the authors to cut out of the paper the work on regions, which is important for the GCAM but a distraction from the main objective of the paper. Instead, the regional work could be the object of a separate paper. This strategy would also allow the authors to target different audiences better.

---

## Author Comment (AC1)

*ESSD: Response to reviewers for Narayan et al. ("A consistent dataset for the net income distribution for 184 countries, aggregated to 32 geographical regions and the world from 1958-2015")*

**Summary of content-**
- **Reviewer 1 responses- Pg 1- 9**
- **Reviewer 2 responses- Pg 9-14**

**Reviewer 1**

**General comments**

*"The paper attempts to create a large dataset of 190 countries over almost 70 years providing consistent information on net income distributions. Such an attempt is valuable. However, if such a database does not already exist, or with limited scope such as the LIS, it is because it raises serious challenges, ultimately related to the lack of suitable data. I found the paper unconvincing in the ways it tackles these challenges. I thus believe that the database it intends to produce (and document) is unlikely to be taken up by other researchers and institutions."*

**Response: Thank you for your detailed review of our manuscript. We have responded to each of your specific comments below. Here, we would like to mention that we have edited the manuscript to better explain the purpose of our dataset construction. In particular, we constructed this dataset to calibrate inequality metrics in regional and global integrated assessment models (IAMs). These models require income distribution data that is comparable across countries. As the reviewer correctly notes, there are surveys conducted in individual countries in individual years, however it is difficult to extract comparable metrics across countries from these datasets for income groups. The LIS and the PovCal are the only two surveys which produce metrics comparable across countries, hence we started with these sources. But even these sources contain a mix of income concepts which is why our imputation (between consumption and net income) was necessary. But, we have clarified that such an imputation is applied to only a small subset of observations in our dataset (394 out of 8522). Finally, models such as GCAM operate at regional scales, some of which aggregate across countries. Thus, we document national income distributions aggregated to the GCAM regional level. Thus far, most IAMs to our knowledge have used income inequality data from LIS and PovCal and have used the different definitions of the income concept interchangeably, hence our dataset is an improvement upon the existing literature. These points have been elaborated upon in our revised manuscript, especially in the introduction and discussion.**

**Specific Comments**

1. *"Imputing net income shares using consumption shares:*
a. *I am skeptical of this approach without more information about the estimation sample and the country-years for which such imputation is performed. If the two sets of countries are different, there are reasons to doubt that good R-squares would translate into good out-of-sample predictions.*
b. *I understand that the current setup with 10 regressions does not ensure that all income shares add up to one. Why not run one regression and impose the constraint that the sum of all income shares must be equal to 1?"*

**Response:**

> a. **This has been addressed in section 2.3 of the paper in detail (We have now provided more detailed information here). In particular,**
> **i.) we used a dataset of 257 country-year observations which had data for both net income and consumption.**
> **ii.) We split this dataset into a training data set (all pre 2004 observations) and a testing dataset (observations starting 2004 onwards) and fit ten separate regressions where we impute individual net income deciles from consumption deciles. See below for the validation of our imputation approach.**
> **iii.) Most of these regressions had an R squared of over 0.6 except the regression for d9 which was 0.29.**
> **iv.) For this reason, we impute net income shares for 9 deciles (all deciles excluding d9) and then calculate d9 as the residual. This resulted in all deciles adding up to 1 for all country-years, which we have verified. We have now made this clear in the text.**
> **v.) If the regression introduced inconsistencies between deciles (e.g., d7 > d8), the GINI coefficient thus calculated would yield an incorrect number. Therefore, we re-calculated a GINI coefficient from our imputed deciles to ensure that there are no inconsistencies between deciles.**
>
> **Validation of fit for our imputation method- To validate our imputation method we calculated errors (Imputed shares - actual shares) for our testing dataset (n=123). We compared the error by decile for the dataset (See Figure below). This figure below is attached as SI 2 Figure 4. The mean error across deciles is generally within half a percent across all years. There are larger differences for the year 2011, where we had very few observations.**

[Figure]

Error from imputation of net income for the pooled testing dataset (*post 2004*). Error unit is in %.

*Figure : Percent Error (imputed income deciles- actual) for the testing dataset. Error is shown for 3 deciles, namely d1, d5 and d10 for all years in the testing dataset .*

**Similarly, we also compared the fit for individual countries from our testing dataset (Figure below). Once again we note that the fit is reasonable across deciles for individual countries. We are able to reproduce the R squared documented from the training dataset in our plot below. Also note that even when R squared values based on the testing data are lower, all imputed values are within a 95 percent confidence interval of actual values. Note that the figure below is attached as Figure 4 in the revised paper.**

[Figure]

*Figure: Comparison of imputed and actual net income decile values for the testing dataset across all deciles. We also show the R squared from the fit here.*

**b.) As noted in point iv.) and v.) above we have noted that our current approach produces deciles that add up to 1. We would also like to note that the imputation affects a small subset of data points (394 out of 8522). The majority of other observations are calculated using the PCA algorithm, whose fit has been clarified in more detail below.**

    *2. "Imputing net income deciles based on summary measures of the Gini coefficient*

*a.  The same point mentioned above about in-sample vs. out-of-sample predictions applies here too.*

*b. Where it is not known whether the Gini coefficients are based on income or consumption, it would be best to drop these countries and years to ensure consistency of the income concept."*

**Response:**

**a.) Thank you for this comment. We introduced the PCA algorithm in Narayan et al. 2023-** **https://iopscience.iop.org/article/10.1088/1748-9326/acbdb0/meta****, where we extensively validated the fit of the algorithm (both in-sample and out-of-sample). We examined the fit for our pooled dataset when compared to other methods (See Figure 1 and Figure 4 of that paper) and we also produced comparisons for individual countries and deciles (See SI Figure 12 and SI Figure 13 of that paper). Our algorithm was found to provide a better fit across all deciles and countries. We have now added more text related to the PCA algorithm fit when applied to our dataset. We could bring in more figures from that paper if useful. We have attached below the main figure from that paper which shows the improved fit for the pooled dataset (The orange dots below represent the fit based on the PCA model)-**

[Figure]

*Figure : Comparison of fit of lognormal functional form (grey dots) with PCA based fit (orange dots) with data for each decile (facet). Lines represent 1 to 1 fit between x and y axis. Income shares are expressed as a percent of total income.*

**b.) Thank you for this very insightful comment. Regarding the concern about using observations when the GINI value is based on consumption data, as noted in section 2.4, we have now identified all observations imputed from a consumption GINI. These have been marked separately in the dataset itself. Users of our dataset have informed us that they would still need a full time series irrespective of the imputation method, hence these observations are still retained. We have updated the table 4 in the paper with the revised**

**statistics regarding the observations. We agree with the reviewer that this is an issue that deserves attention and could be revisited in the future.**

| Data source | count of observations |
|---|---|
| Original data on net income | 1191 |
| Imputed from original data on consumption | 494 |
| Imputed from Net income GINI coefficient | 4201 |
| Imputed from Expenditure & Consumption GINI coefficient | 1303 |
| Imputed from Gross income GINI coefficient | 1333 |
| **Total** | **8522** |

*Table* : *Summary of observation types in final data set*

*3. "Aggregating income distributions to the regional level*
*a.) I do not see a clear motivation for this section (other than the need for the authors to carry out these analyses for another project/report).*
*b.) The approach sounds highly problematic as it appears to confuse (or ignore the differences) between household net income and GDP per capita. In addition, it also appears to ignore (crucial) variations in income dispersion within income decile groups.*
*c.) The same issues apply to section 4, in which the authors aggregate country income distributions up to the global level."*

**Response:**

**a.) Thank you for this comment. We have clarified in more detail why this step was necessary. In particular, we constructed this dataset to calibrate inequality metrics in regional and global Integrated Assessment Models (IAMs) such as GCAM. Regional models such as GCAM require regional boundary conditions, thus we developed a method to aggregate national income distributions to GCAM regional levels as an example. If national data on income distribution was to be used in any other model , such an aggregation method would be necessary. This has been clarified in section 3 of the manuscript now.**

**b.) Thanks again for the comment. We agree that our method for aggregation is subject to uncertainties and limitations. Firstly, regional economic models use GDP per capita as a proxy for income levels, hence we used the same variable to perform the aggregation. We**

can alternatively use net income; however, we wanted to be consistent with the model's measure of income. We have noted that our current method ignores within-decile variations in income levels and assumes a uniform distribution of income within a decile. This can and should be improved upon as more data on income distribution become available. We have clarified this in our manuscript.

c.) As noted in point b.) above, we agree that our method is subject to limitations. We agree with the reviewer that aggregation to the global level would introduce more uncertainties and have dropped this section from the manuscript. We have also edited the title of the paper.

*4. "The differences between the different data sources shown in Figure 5 are concerning. Given these sizeable differences, it is far from clear that one could accurately assess inequality levels and trends using data imputed by the authors."*

Response: The largest difference here is noted for the US. As noted in the previous comment, this is because we used the GINI data based on gross income to estimate inequality in the US when no data was available. This is because the ACS data, which produces data on income distributions in the US, is based on gross income as opposed to net income. We have dropped the observations based on the gross income GINI when constructing the figure. When we re-made the figure solely based on net income observations or those imputed from a net income GINI, this solved our issue as noted in figure 5, also attached below. Note that there can still be jumps between years (For example d10 income shares in India by 7% between 2005 and 2010) but this is possible since the survey design can change between years and between data sources themselves in different years. However, with our method, jumps between years are limited.

[Figure]

*Figure : Temporal trends in the 10th decile for the complete dataset. Colors represent different data sources.*

5. "*US results (Fig 5): why is "original data" for net income not available for the whole period? The CPS is a large and representative survey that collects detailed income information and that has been running yearly since the 1960s.*"

**Response:  Thanks for this comment. As the reviewer correctly notes, there are surveys conducted in individual countries in individual years (such as the CPS), however it is difficult to extract comparable metrics across countries from these datasets for income groups. The LIS and the PovCal are the only two datasets which produce metrics comparable across countries, hence we started with these sources. The CPS is based on consumer expenditures, and its Annual Social and Economic Supplement (ASEC) records only gross income. Neither produce the income concept that we are interested in, namely net income. We have clarified this point throughout our manuscript.**

6. "*Introduction : the paragraph starting on line 29 is odd because it suggests that "at the national level", datasets on income inequality have been "limited to summary metrics". That is clearly not true. In many countries, detailed microdata allows researchers and statistical agencies to produce detailed distributional analyses.*"

**Response: As the reviewer correctly notes, there are surveys conducted in individual countries in individual years which produce very useful microdata, however it is difficult to extract comparable metrics across countries from this microdata for income groups. The LIS and the PovCal are the only two datasets which produce metrics comparable across countries, hence we started with these sources. We agree that microdata is available across countries and have acknowledged that in our manuscript now.**

**Reviewer 2**

1. *"The paper contributes to a growing literature on world databases of incomes. The data sources selected are appropriate but insufficient. There is no mention of the work led by Branko Milanovic in the US and Thomas Piketty in France. Both these authors oversee global data efforts to measure income and income inequality that should be reviewed/acknowledged by the authors."*

**Response: Thank you for the careful review of our manuscript and your comments. As noted in the responses to reviewer one, we have added text in section 2.3 to address these points. We have acknowledged the Milanovic and Piketty approaches in our citations and text and agree that it is important to highlight such efforts in this space. However, we have noted that the Lanker-Milanovic dataset is still a combination of PovCal and LIS and has limited temporal coverage (the data is only available to the year 2013). The Piketty dataset is available only for the USA and is not global.**

**We have not drawn on these data directly because we constructed this dataset to calibrate inequality metrics in regional and global economic models. These models require income distribution data that is comparable across countries i.e. the same income concept.**

2. *"The central contributions of the paper are the regression-based approach to estimate net incomes from consumption data and the PCA approach used to estimate income data from national Gini coefficients. The regression-based approach (section 2.3) is popular but there are better methods that should provide better fit including quantile regressions and random forest methods. The problem with simple OLS is that they are very poor at predicting incomes on the tails of a distribution (the predicted income distribution is always much narrower than the original income distribution). The fact that the authors run different regressions for each decile accentuates this problem by creating discontinuities between deciles. This problem can be overcome with quantile regressions or, better, with random forest. The probabilistic nature of random forest fits the tails if income distributions much better than standard OLS. I would recommend the authors to test both methods and compare results with the current ones."*

**Response: Thank you for this very thoughtful comment. We agree that the linear regression we implemented for our imputation is simplistic. However, we justify its current usage based on several points,**

**a.) We first note that the imputation affects a small subset of data points (394 out of 8522). The majority of other observations are calculated using the PCA algorithm, whose fit has been clarified in more detail in the paper (as we also describe below).**

**b.) We have now described our approach for imputation in more detail and added new validation information, namely-**

> **i.) we used a dataset of 257 country-year observations which had data for both net income and consumption.**
> **ii.) We split this dataset into a training data set (all pre-2004 observations) and a testing dataset (observations from 2004 onwards) and fit ten separate regressions where we impute individual net income deciles from consumption deciles.**
> **iii.) Most of these regressions had an R squared of over 0.6 except the regression for d9 which was 0.29.**
> **iv.) We impute net income shares for 9 deciles (all deciles excluding d9) and then calculate d9 as the residual. This resulted in all our imputed deciles adding up to 1.**
> **v.) We re-calculated a GINI coefficient from our imputed deciles to ensure that there are no inconsistencies between deciles. Note that if the regressions introduced inconsistencies (e.g., if imputed d7 is higher than imputed d8), the GINI coefficient calculation would result in implausible values.**
> **c.) We also performed several types of validation for our imputation-**

> **To validate our imputation method, we calculated errors (Imputed shares- actual shares) for our testing dataset (n=123). We compared the error by decile for the dataset (See Figure 1 below). The mean error across deciles is generally close to zero across all years. There are larger differences for the year 2011, where we have very few observations.**

[Figure]

Error from imputation of net income for the pooled testing dataset (*post 2004*). Error unit is in %.

*Figure : Percent Error (imputed income deciles- actual) for the testing dataset. Error is shown for 3 deciles, namely d1, d5 and d10 for all years in the testing dataset .*

   Similarly, we also compared the fit for individual countries from our testing dataset (Figure 2). Once again we note that the fit is reasonable across deciles for individual countries. The Figures shown here are attached in the revised paper as SI 2 Figure 4 and Figure 4 respectively.

[Figure]

**Comparison of Distribution values (using imputed income shares) across countries, years from the testing dataset**

n = 109 , Units are in percent (percent of total income)

*Figure: Comparison of imputed and actual net income decile values for the testing dataset across all deciles. We also show the R squared from the fit here.*

3. *"The PCA method (section 3.4) is somewhat unconventional for this specific literature. This, per se, is not a critique, but it does require validation beyond what the authors offer. Here I would suggest taking the entire net income distribution for a few countries where these data are publicly available, calculate the net income deciles and the Gini coefficient, plug this Gini into equations 3) and 4) and compare the resulting estimated deciles with those calculated from the full data. Also, it is important to clarify where the coefficients in equations 3 and 4 come from. I could not find the model and the results of the "equation estimated on 1659 observations".*

**Response: We introduced the PCA algorithm in Narayan et al. 2023-**
**https://iopscience.iop.org/article/10.1088/1748-9326/acbdb0/meta, where we extensively**
**validated the fit of the algorithm. We examined the fit for our pooled dataset when compared**
**to other methods (See Figure 1 and Figure 4 in that paper) and we also produced comparisons**
**for individual countries and deciles (See SI Figure 12 and SI Figure 13 in that paper). Our**
**algorithm was found to provide a better fit across all deciles and countries (See Figure**
**attached below). We have now added more text related to the PCA algorithm fit for our**
**dataset in this paper. We can bring in more figures from our older paper here as well, but we**
**leave that to the discretion of the editor.**

[Figure]

*Figure : Comparison of fit of lognormal functional form (grey dots) with PCA based fit (orange*
*dots) with data for each decile (facet). Lines represent 1 to 1 fit between x and y axis. Income*
*shares are expressed as a percent of total income.*

4. *"The revisions suggested above are substantial and I would recommend the authors to cut out of*
   *the paper the work on regions, which is important for the GCAM but a distraction from the main*
   *objective of the paper. Instead, the regional work could be the object of a separate paper. This*
   *strategy would also allow the authors to target different audiences better."*

   **Response: Thank you for this comment. We have clarified in more detail why this step was**
   **necessary. In particular, we constructed this dataset to calibrate inequality metrics in**
   **regional and global economic models such as GCAM. Regional models such as GCAM**
   **operate on regional boundary conditions and hence it was necessary to produce a method**
   **to aggregate national income distributions to the regional level. This ensures that models**
   **can be effectively calibrated. If national data on income distribution was to be used in any**
   **other model, such an aggregation method would be necessary. This has been clarified in**

**section 3 of the manuscript now. Also, we removed the section of the paper on the aggregation to the global level, as regional models do not need to do such an aggregation. We also responded in more detail to this point based on comments from Reviewer 1.**

---

## Author Response (AR2)

*ESSD: Response to reviewers for Narayan et al. ("A consistent dataset for the net income distribution for 184 countries, aggregated to 32 geographical regions and the world from 1958-2015")*

**(Revisions for Round 2)**

**Summary of content-**
- **Reviewer 1 responses- Pg 1- 6**

**Reviewer 1**

**General comments**
*"The paper describes an empirical dataset on a large set of countries and time periods covered of household deciles, which are a very useful exercise.*
*Overall it is well written, and the produced data looks indeed very good and addresses several issues in existing data sets. However, I have a few concerns on the presentation and questions about few references that would be good to take into account before publication."*

**Response: Thank you for your detailed review of our manuscript. We have made several edits in response to your comments. In particular, we added more details with respect to data coverage, and have documented several points in the methodology section more clearly. The responses to the specific comments are attached below. Note that we have mentioned specific sections with page numbers, line numbers where edits are made.**

**Specific Comments**
1. *"Data coverage: Table 6 is not very informative (average populations of countries not very useful, maybe total population/GDP covered or not would be more meaningful, best in a plot or map). And what are the biggest countries missing? Secondly, YEARs: Why does you r data series stop in 2015? In 2023 this is a big issue for using the data for empirical researchers, is there a reason? Finally, a heat map of all years and countries would be nice. Is it a panel without gaps ultimately?"*
**Response: Thank you for the insightful comments. In response to the same,**

   **i.)**      **We have edited Table 6 to represent the % of GDP and population covered. We utilized the GDP and population data from the SSP database for the same. The same is attached below. As seen in the table a majority of the global population and global GDP is covered in our dataset. We also added the % of countries that are imputed from the GINI (using the PCA algorithm) and the ones imputed from consumption (using the regression equation).**

| Country data status | year | Global GDP\|PPP | Global Population |
|---|---|---|---|
| **Data not available** | **2010** | **0.4%** | **2.0%** |
| **Data not available** | **2015** | **0.3%** | **1.3%** |
| Imputed from GINI coefficient (using PCA algorithm) | 2010 | 19.9% | 25.8% |
| Imputed from GINI coefficient (using PCA algorithm) | 2015 | 45.1% | 52.5% |
| Imputed from original data on consumption (Using regression) | 2010 | 10.8% | 31.2% |
| Imputed from original data on consumption(Using regression) | 2015 | 5.8% | 9.6% |
| Original data on net income | 2010 | 68.9% | 41.0% |
| Original data on net income | 2015 | 48.8% | 36.6% |

*Table 6: Coverage by data status in terms of GDP in PPP and Population from the SSP database V9.*

ii.) **We have also noted the major countries that are missing in section 4 (Line 2-7 on Page 22 of the revised manuscript). Specifically, we note "** *found that the countries that are missing data in the latest historical year (2015) only constitute 1.3% of the global population and 0.3% of the global GDP. The biggest countries that are missing data in terms of population in 2015 are Morocco (33 million people), North Korea (24 million people) and Somalia (10 million people). In terms of GDP, the biggest countries missing are Morocco ( 123 billion USD at PPP), Oman (68 billion USD at PPP) and Equatorial Guinea (18 billion USD at PPP)***"**

iii.) **In addition to the edits to Table 6, we also show a map of countries by data status in Figure 11 (attached below). This allows the user to understand countries that are missing recent data on income distributions. We further added all categories in our final dataset to the map below.**

[Figure]

*Figure 1:Data availability by country. Availability is shown here for two years 2010 and 2015.*

    **iv.)**     **We stopped our dataset in 2015 since that is the last historical year currently in the GCAM model. We can and should extend this dataset to other years in the near future as more data become available. This is now noted in the discussion section of the revised manuscript. We have made our code available, so this update should be straightforward.**

    **v.)**     **Our final dataset is a panel without gaps. For each year, we note where the data came from (original data vs imputed data from consumption vs imputed data from GINIs)**

*2. "The clear distinction of net, gross income, and consumption is a great contribution of this article. but not sure if I would call it "income concept" as consumption is not income. Rather a concept of inequality, income or consumption. But just my thought here."*

  **Response: We agree with the reviewer regarding this point. We would note that the references to the "income concept" are available in the literature, especially the cited papers such as Deaton & Zaidi (2002). However, we have edited the abstract (Line 12-13) to introduce the "concept of inequality (or income concept)". Similarly, we have defined this as**

**the concept of inequality but mentioned that we will call it an income concept for convenience up front (on line 4-5 Page 2).**

*3. "Three notably sources of inequality data I see not references here, so a paragraph on those or why or why not they are included would be good to complete the picture: The SWIID database, the WIID database (which might have been used, but the name is not given), and the WID by Chancel et al. (All three databases can be easily found online)."*

**Response: We have now addressed this in section 2.1 (Literature review) on lines 1-8 Page 5. We are primarily interested in decile-level income distributions derived from household surveys. Given our criteria for data selection, we limited our data collection to the datasets mentioned in the manuscript (LIS, PovCal). We did not use the Standardized World Income Inequality Database (Solt, 2020) since it includes only the GINI coefficient and not a full distribution by income groups (such as deciles). Moreover, the concept of inequality in this dataset is disposable income or market income as opposed to net-income. Similarly, we did not use the World Inequality Database (Chancel et al 2021) since this dataset is not based on household survey data (This database uses a distributed national account methodology). Finally, we confirm that we have used the WIID database. We refer to it as the UNU-WIDER database. This is now clarified in the manuscript.**

*4. "Equivalence scale: this is an important feature of income and inequality measurements. Some more detail would be good. You argue on page 6 to use "same per-capita equivalence scale". First, it is not clear what this means, but you seem to divide HH income by all members, counting even children by 1. Are many surveys not use say the OECD modified or square root or similar equivalence scales? This is an important issue in my opinion to at least discuss."*

**Response: Thank you for this comment. We have clarified in the manuscript (Page 8, Line 9-11), that we selected observations that were all represented at the same per capita equivalence scale. The UNU WIDER dataset presents data across sources at the same scale and in fact allows users to select from amongst 3-4 options. We selected the per capita scale since it was the most appropriate for GCAM. We have not tested the effect of changing the equivalence scale (e.g., OECD vs per capita) on the income distribution, however. This can be explored separately in future studies.**

*5. "Sections 2.x should be slightly improved in terms of presentation. In 2.3 the regression estimation would be good. The sentence "We calculate values for 9 deciles d1-d8 and d10 and the re-calculate d9 as the residual." needs to be explained. Why d9 as residual? Equation 1 is a projection but should be a regression estimation in my opinion. And please harmonize notation with the PCA section. Would suggest beta for "Coeff". etc.."*

**Response: Thank you for this comment. We have made several corrections to this section. Namely-**

    **i.)**      **The notation of equation (1) has been changed and the parameters are presented similar to the PCA equations later in the manuscript (Page 12).**

    **ii.)**      **We have clarified that this equation is a regression equation (Line 6-7 Page 11). This has also been clarified by moving Table 4 up so that it is clear that we used 10 separate equations for each decile.**

    **iii.)**    **Finally, we have clarified that d9 is calculated as the residual since it had the lowest R squared (0.43) amongst the 10 regression equations for the deciles (Line 1-2, Page 11).**

6. *"You refer to another paper of yours (Narayan et al. (2023)) describing the method of PCA in detail. But that paper seems not to apply to historical but future scenario data, correct?"*

**Response: Yes. This PCA model was indeed used to generate future projections. However, this was fit to the latest available data on net-income distributions (only original data with no imputation) and was found to provide a better fit compared to other imputation methods such as the lognormal functional form (See Figure 5 in the manuscript also attached below). Hence this method was used to impute historical values where only the GINI is available. This is now made clear in the Introduction (Line 11-13 Page 3) and in section 2.4 (Line 22-25 Page 14).**

[Figure]

*Figure 2: Comparison of fit of lognormal functional form (grey dots) with PCA based fit (orange dots) with data for each decile (facet). Lines represent 1 to 1 fit between x and y axis. Income shares are expressed as a percent of total income.*

7. *"One bis issue with inequality based on survey data is that surveys in many countries are run only every x years, so the time gap is one of the big issues, yet this is not mentioned specifically in the paper, might be good to add. And related to this, in the cases where an aggregate inequality index was available, how had that been computed, as there must be some underlying distribution and hence deciles available."*

**Response: Thank you for this very thoughtful comment. We have not addressed the issue of variability of surveys between years. However, the LIS and the PovCal have consistent methods in each year which is a big advantage of this data. We used the WDI GINI to fill in holes. But we have noted that the WDI does not provide any details on underlying surveys. As a first step, we used the All the GINIs dataset to at least understand when a GINI is based on consumption, gross income or net income (These points are documented in section 2.4 now, on lines 13-16 Page 17). But as more data become available, this should be addressed more directly. We have added these points to the discussion.**

---

## Author Response (AR3)

*ESSD: Response to reviewers for Narayan et al. ("A consistent dataset for the net income distribution for 184 countries, aggregated to 32 geographical regions and the world from 1958-2015")*

**(Revisions for Round 3)**

**Summary of content-**
- **Reviewer 1, Editor responses- Pg 1**

**Reviewer 1**

**General comments**
*"Thank you for considering the points raised in the last version, which are satisfactorily addressed.*
*Only the relevance of a single model's base year does not strike me as a good reason to limit coverage notably temporarily. That is, if possible, I would applaud if the authors could expand their dataset to the early 2020s.*

*Lastly, I would improve the layout of tables considerably which are hard to read and formatted unfortunately."*

**Response: Thank you for the positive comments on our manuscript. In response to the comments, we have added a section 5 entitled " Updated data in the future" where we discuss how the dataset can be updated in the future by users. We have also added a link to the code that was used to generate the data. We have also re-edited the tables to make sure they are correctly presented. We also note that we intend to update the data to the early 2020s and are in the process. We realized that extending the data would require more data compilation which may take more time to complete than earlier envisioned. Thank you again for the comments.**